# Smoking and Vaping in Amateur Rugby Players, Coaches and Referees: Findings from a Regional Survey Might Help to Define Prevention Targets

**DOI:** 10.3390/ijerph18115720

**Published:** 2021-05-26

**Authors:** Frédéric Chagué, Emmanuel Reboursière, Jean Israël, Jean-Philippe Hager, Patrice Ngassa, Marc Geneste, Jean-Pierre Guinoiseau, Gilles Garet, Jacques Girardin, Jacques Sarda, Yves Cottin, Marianne Zeller

**Affiliations:** 1Cardiology Department, University Hospital Center Dijon Bourgogne, 21000 Dijon, France; yves.cottin@chu-dijon.fr; 2French Rugby Federation, 91460 Marcoussis, France; emmanuel_reboursiere@hotmail.com (E.R.); jeanrichardhenri@gmail.com (J.I.); hager.md@orthosanty.fr (J.-P.H.); ngassa.patrice@wanadoo.fr (P.N.); mgeneste@wanadoo.fr (M.G.); jean-pierre.guinoiseau@ffr.fr (J.-P.G.); gilles.garet@wanadoo.fr (G.G.); jacques.girardin@ffr.fr (J.G.); sardajacques@yahoo.fr (J.S.); 3Sport Medicine Department, University Hospital Center, 14000 Caen, France; 4Cardiology Department, General Hospital, 91640 Bligny, France; 5Sport Medicine Department, Centre Orthopédique Santy, 69008 Lyon, France; 6Sport Medicine Department, Polyclinique Saint-Laurent, 35700 Rennes, France; 7Physiopathologie et Epidémiologie Cérébro-Cardiovasculaires (PEC2), EA 7460 UFR Sciences de Santé, University of Bourgogne Franche Comté, 21000 Dijon, France; marianne.zeller@u-bourgogne.fr

**Keywords:** tobacco, nicotine, cardiovascular risk, rugby, sport, electronic cigarettes

## Abstract

A high prevalence of cigarette smoking has been documented in France, and new patterns of tobacco and nicotine consumption are emerging, especially in some sports. In the amateur rugby population, data are scarce on harmful consumption and on the awareness of the risk of smoking. We analyzed the consumption of tobacco and other nicotine products in French amateur players, coaches and referees. Amateur players (>12 years old), coaches and referees participating in the Burgundy amateur championship were invited to answer an electronic, anonymous questionnaire during the 2017–2018 sport season. Among the 683 subjects (gender ratio M/F = 0.9), 176 (25.8%) were current smokers, including 32.4% of the referees and 28.2% of the coaches. The prevalence of smokers was higher in females (37.5%) than in males (24.6%). Most (86.4%) smoked within 2 h before/after a rugby session. Only 28 subjects (4.1%) usually vaped; 21 of them (75%) vaped within 2 h before/after a rugby session. Other tobacco or nicotine products were infrequent. The awareness about the risks of smoking before/after sport was incomplete, including in coaches and referees. The prevalence of cigarette smoking is alarming across the whole spectrum of rugby amateur actors. Education programs are urgently needed to reduce tobacco consumption in this at-risk population.

## 1. Introduction

Tobacco smoking and physical inactivity, which are key avoidable risk factors for cardiovascular (CV) disease, strongly compromise health and life expectancy worldwide [1,2].

In France, although a substantial decrease has recently been observed, the rate of tobacco smoking remains high; in 2017, the rate of daily smokers was 26.9% among the 18–75-year-old French population [3]. Other forms of tobacco or nicotine consumption such as smokeless tobacco, waterpipes and electronic cigarettes (e-cigarettes) are currently emerging, which may also increase the risk of CV disease [4,5,6,7]. In France, the main reason to use e-cigarettes is to quit smoking. A drop in the French vaping rate in individuals aged 15–75 years was observed between 2014 and 2016 (5.9% to 3.3%, respectively). In contrast, a huge increase was seen in US high school students between 2011 and 2018, from 1.5% to 20.8% [5,8].

Tobacco smoking is a major risk factor for acute coronary events, which are the main cause of sudden cardiac arrest during exercise [9]. Smoking impairs physical capacity through many pathways including early vascular damage [10]. Acute and chronic e-cigarette exposure has been associated with vascular dysfunction in recent preclinical and clinical studies [7,11]. Tobacco must be strongly discouraged because it promotes acute CV events, in particular when consumed shortly (within 2 h) before or after physical activity [12]. Over the last decade, tobacco consumption has rarely been explored in athletes, and there is a paucity of contemporary data regarding alternative consumption patterns such as smokeless tobacco or e-cigarettes in France [13,14,15,16,17]. We previously documented an alarming rate of smokers among amateur rugby players, and this was observed in both genders; however, there were no data concerning other types of nicotine consumption [17]. Awareness of the risks of smoking and the smoking behaviors of coaches and referees remained unknown, and their social and educational role may confer a point of interest when considering their influence on players [18]. We aimed to investigate tobacco and e-cigarette consumption patterns, behaviors and knowledge across the spectrum of French amateur rugby, including players, referees and coaches, on a regional scale.

## 2. Methods

From among the players in the 2017–2018 French amateur rugby championship from the Burgundy region, 5547 subjects were invited to participate, including 4978 players, 1184 coaches and 84 referees (of note, some subjects had simultaneous roles).

The 5547 subjects {4589 males (82.7%)} represent 0.34% of the Burgundy region population {1,632,887 inhabitants in 2017, 790,673 males (48.4%)}. The Burgundy League includes 59 rugby clubs, of which one club is involved in professional championships and was excluded from the survey (55 players). There is not an upper age limit to play in the league, although players >55 years old are rare. Coaches and referees must be more than 18 years old, and referees cannot be older than 55 years old.

A link inviting players, coaches, and referees of the Burgundy amateur rugby championship was displayed on the Burgundy rugby committee website from 15 October 2017 to 15 May 2018. Each club manager received three successive emails (October 2017, January, and March 2018) to motivate their players, coaches and referees to participate to the study. Data were confidentially collected.

The study questionnaire included 21 items (translated questionnaire attached as a Appendix A) that addressed the following three subjects:

(1) Consumption patterns for combustible cigarettes, e-cigarettes, and other derived products, with a special focus on consumption in a sports setting (i.e., within 2 h before and/or after rugby session, training, or match). Smoking status was defined as current cigarette smoking or a cessation of smoking within the prior 3 months;

(2) Perception of self-consumption and strategies for quitting;

(3) Knowledge of the risks to CV health linked to smoking or vaping in general and to a sports setting specifically.

Selected items were created for the questionnaire and tested in 20 subjects (2 females). Questions generated single (yes/no) or multiple answers according to the type of item; none of them used a scale.

The present study complied with the Declaration of Helsinki and was approved by the Ethics Committee of the Dijon University Hospital for the whole population including minor subjects.

### Statistical Analysis

Continuous variables are reported as means and standard deviations (SD) or medians and interquartile ranges (IQR), when appropriate. Discrete variables are described as counts and percentages and were compared with chi-square or Fisher’s exact tests. Statistical difference was defined as a *p* value < 0.05. All analyses were performed using SPSS 12.0 software package (IBM Inc., New York, NY, USA).

## 3. Results

Among the 5547 players, referees, and coaches, 683 (response rate = 13.6%) completed the survey; most participants were male {*n* = 619 (90.6%)} (Table 1).

Most respondents were players {*n* = 559 (81.8%)}, followed by coaches {167 (24.5%)} and referees {74 (10.8%)}. Demographic data of respondents are summarized in Table 1. Some subjects belonged to multiple categories (Figure 1). When compared with the whole rugby-licensed population (*n* = 5547), respondents were similar in terms of gender (rate of male = 90.9% vs. 90.6%, respectively; *p* = ns) and for most age groups, except for subjects <18 y who were slightly underrepresented (24.9% vs. 40.1%, respectively; *p* < 0.001), suggesting the representativeness of the respondents. The response rates were much higher in referees than in coaches and players (88.1%, 14.1%, and 11.9%, respectively), (*p* = ns players vs. coaches; *p* < 0.001 players vs. referees and coaches vs. referees).

### 3.1. Consumption of Combustible Cigarettes, e-Cigarettes and Nicotine Containing Products

Among the study population, 176 were current combustible cigarette smokers (25.8%) and just less than half (46.9%) had never smoked. In the whole group and in the players, smoking prevalence was higher in females than in males (37.5% and 38.6% vs. 24.6% and 23.9%; *p* = 0.034 and 0.023, respectively) (Figure 2). There was only a trend toward a higher rate of smoking in females among coaches (*p* = 0.509). Smoking prevalence was highest in the 20–24 and 25–29-year-old age groups, with significant differences when compared with that of younger respondents (i.e., 15–19-year-old age group) (*p* = 0.008 and 0.006, respectively) (Table 2; Figure 3). Among smokers, most {*n* = 126 (71.5%)} declared daily tobacco consumption and 103 (58.5%) declared a mean consumption of >5 cigarettes per day. Some subjects {97 (14.2%)} were former regular smokers, 19 (19.6%) of whom had quit within the last year. Among the 109 (61.9%) smokers who declared intending to quit, almost a quarter {*n* = 27 (24.8%)} planned to use e-cigarettes as a quitting strategy (Table 3).

Among smokers, the rate in daily smokers, number of consumed cigarettes, smoking behavior in a sports setting, rate of intention to quit, and intention to quit using electronic cigarettes did not differ according to gender.

The rate of tobacco use in players was unchanged if they had an additional role (i.e., coach and/or referee) (30.4% and 24.3%, respectively; *p* = 0.20).

Experience with e-cigarettes was reported by 343 (35.6%) subjects, and 28 (4.1%) subjects were currently vaping (Table 2). Prevalence was higher in the 25–29 age group (*p* = 0.04) and tended to be higher in females (*p* = 0.32). Younger females (15–19 y) had the highest rate of e-cigarette use (13%). The majority used nicotine-containing cartridges {*n* = 22 (81.5%)}. Only 17 respondents were former e-cigarette users (2.5%), 6 (35.3%) of whom had quit within the last year. Among the 17 former e-cigarette users, 6 were also former combustible cigarette smokers, and 11 were still smoking. The most common reasons for e-cigarette use were to quit smoking {11/28 (39.3%) vapers} and to avoid smoking {5/28 (17.9%) vapers}. The use of e-cigarettes as a device to avoid the effects on performance of smoking combustible cigarettes was reported by only 3 e-cigarette users. Among the e-cigarette users, 14/28 (50.0%) preferred vaping rather than smoking because it was less hazardous for their health. One late-teens player declared vaping on occasion but with no previous experience of smoking, and another (early twenties) declared vaping daily but having tried smoking only a few times. Overall, most e-cigarette users (19/28; 67.9%) were dual users (Table 4). Their combustible cigarette consumption was similar to that of exclusive smokers (Figure 4). The intention to quit was declared by 13/27 vapers (48.1%).

Other tobacco products were rarely consumed (15; 2.2%). Waterpipes were the most common (7; 46.6%). Four respondents smoked cigars, one used dry snuff, and none reported using snus.

### 3.2. Consumption of Tobacco Products in the Setting of a Rugby Session

Smokers declared combustible cigarette smoking in a sports setting, i.e., within 2 h hours before and/or after a sport session (training or match) (Figure 5). Most subjects smoked before a training session (*n* = 130; 73.9% of smokers) or a match (*n* = 110; 62.5% of smokers). The proportion was even higher after sport, with 140 subjects smoking after training (79.5% of smokers) and 142 after a match (80.7%). Only a few smokers (*n* = 24; 13.6%) never smoked within the two hours preceding or following a sport session. In this context, no difference was observed between players, coaches, and referees. The rate of smoking before/after the rugby session was similar for both genders (male: 75.0% vs. female: 80%, *p* = 0.748), but there was a non-significant trend toward a higher rate among older subjects (>35 years) than younger subjects (<35 years) (91.4% vs. 85.2%, *p* = 0.35). This increased rate was only significant in the 15–19 vs. 35–39 year age groups (71% vs. 100%, *p* = 0.021).

In a sports setting, most vapers (21/28; 75%) reported e-cigarette consumption within the two hours preceding or following a sport session (Figure 5). Thirteen of these vapers were players, four of whom vaped before a match. One vaper was a dual user who did not smoke before a match. Among dual users, only 2/19 neither smoked nor vaped before and/or after a training session or a match.

### 3.3. Risk Perception of Tobacco Product Use in a Sports Setting

Self-perception of the risk for cigarette smoking and vaping is reported in Figure 5. When the question addressed the age-related health risks of smoking, three smokers thought that their own risk was low and five did not know.

The risk linked to smoking within the two hours before and after the sport-session period was unknown by 291 (42.6%) and 213 subjects (31.2%), respectively. Only 367 (53.7%) were aware of the danger of smoking before sport or after sport (Figure 5). Smokers were more aware of the risk of smoking before or after exercise than non-smokers (80.1% vs. 69.8%, *p* = 0.008). The level of awareness was similar for both genders in the study population (*p* = 0.513) and among smokers (*p* = 1). Awareness of at-risk behaviors (smoking before/after session) was greater for smokers (116/176; 65.9%) than for non-smokers (251/507; 49.5%), never smokers (162/320; 50.6%), never (or almost-never) smokers (204/411; 49.6%) and ex-smokers (47/97; 50.5%), (*p* = 0.0002, *p* = 0.0003, and *p* = 0.0067, respectively). It should be noted that smokers who were aware of the risk were less likely to smoke before or after a rugby session (81.9 and 95.0%, respectively; *p* = 0.019).

We also observed that most coaches (106/167, 63.5%) and referees (46/74, 62.2%) were aware that smoking before and after sport is a risky behavior. Among players, the rate of awareness was lower (294/559; 52.8%) than in coaches (*p* = 0.013) or referees (*p* = 0.137) (Figure 5).

For vaping, three quarters of vapers used e-cigarettes before or after a session. In the whole population, less than 40% (37.6%) were aware that it could be dangerous, which is lower than the proportion of respondents who were aware of the danger of smoking before and after exercise (*p* < 0.001).

## 4. Discussion

Our findings, which were obtained from a survey representative of amateur rugby participants on a regional scale, highlight that combustible cigarette smoking is highly prevalent among participants (players, coaches, and referees). Although females were scarce in our population (<10%), they smoked at a strikingly higher rate than did males. The highest rate, reaching almost 50%, was observed in females aged 20–24 years, and this finding was further associated with engagement in at-risk practices (i.e., smoking before/after a rugby session). In addition, we found that there was a low awareness of the health risks in all participant categories, even referees and coaches.

The prevalence of combustible cigarette smoking (26%) was very similar to the rate found in the French general population aged 18–75 years (25.4%) in 2018, and was consistent across age groups [3]. However, we found a higher prevalence in females than in males. In contrast with the findings from the general population, similar trends were found in rugby players and in various sports [3,14,17]. However, a recent survey conducted in Ireland on 546 predominantly amateur athletes documented a significantly higher prevalence of smoking in male than in female players (20% and 6%, respectively); among the 129 rugby players, the smoking rate was remarkably similar to that of our rugby population (26%) and less similar to the prevalence in the 15–75-year-old general Irish population (22%) [19]. The prevalence of smoking found here is higher than in professional footballers [13]. Smoking rates are lower in elite athletes [13,20], but even so, the rates are higher in high-contact sports than in non-contact sports [21]. The lower prevalence in French college students might be explained by socioeconomic profiles [22].

In a 2014–2015 study among amateur rugby players in the same geographic area, we found that 34.7% of participants were smokers, which is much higher than in the present study [17]. The observed decrease is in line with a recent reduction in smoking rates nationwide [3]. In contrast, we did not find a significant decrease in smoking prevalence in female smokers over the two study periods: 39.4% in 2014–2015 to 37.5% in 2017–2018. The high rate of smoking in female amateur women might be driven by normative behaviors, which are a determinant in sport [21].

We also found that more than one third of participants had experience with e-cigarettes, and a few (<5%) were current users. As for combustible cigarettes, there was a trend toward a higher prevalence of e-cigarette use in females than in males. Experience with vaping (35.6%) was much higher than in the general population of adults or students in France (23% and 24.5%, respectively). The rate of vaping in our population (4.1%) is within the range of current nationwide data (3.3–5.7% and lower than in an Irish sport population (2%) [8,19,22]. Of note, 2.8% of respondents were dual users (i.e., smoking and vaping), but most vapers (65.7%) were also current smokers. In an Irish sport population, some found a higher rate of vapers in current smokers than in non-smokers [19]. Meanwhile, the prevalence of smoking remained stable at around 35% in France in individuals aged 15–75 years [8]. In the US in 2017, the rate of current smokers was around 18% in adults and 8% in youths (15–17 years) [23]. In 4450 US students, the rate of vaping (over the last 30 days), was as high as 18.0%, which was even higher than for smoking (12.1%), and dual users at 7.2%. Smoking and dual consumption were less frequent in the 64.7% of students who were involved in a competitive sport. However, the proportion of vapers remained similar to that of the non-competitive group [24]. In young vapers with no or little experience of smoking (*n* = 2 in our study), e-cigarettes could be a gateway to smoking [11,22]. We did not find an inverse relationship between e-cigarette use and the extent of smoking, in agreement with current reports [22]. The reasons most cited for vaping were to lessen the impact on health and to avoid or quit smoking. Maintaining performance through nicotine delivery was an infrequent reason for vaping [25]. The health risks associated with e-cigarettes were considered to be lower than those with combustible cigarettes. However, recent work supports the hypothesis that there is a significant CV risk associated with e-cigarette use, in particular in dual users [6,11].

Few respondents (2.2%) reported using other tobacco products. Except in regions practicing winter sports such as skiing or ice-hockey, smokeless tobacco consumption is rare in France. Our regional data are also consistent with previous findings, showing a lack of such use [15,17]. Waterpipes, whose popularity is emergent in French youth [8], were reported by only 1% of our subjects in a younger age range (16 to 37 years). The harmful effects of waterpipe use on CV health have been widely described, in particular among athletes [5].

A large number of smokers in our population (>60%) wanted to quit, with a trend toward higher rates in referees than in coaches and players. Among them, only a quarter planned to use e-cigarettes as an aid to quit. A very similar rate (56.5%) of the French population (aged 18–75 years) wished to quit [3]. E-cigarettes have been proposed as an aid for smoking cessation in prevention guidelines [1], and we found this motivation in vapers, similar to a recent study in students [22].

We further addressed smoking and vaping in the 2 h before or after a match. The vast majority of users tended to smoke combustible cigarette (86.4%) or vape e-cigarettes (100%) during this period, irrespective of their age or role. These findings are comparable to previous work [17]. We expected that dual users would switch to e-cigarettes before playing a match to maintain or even to enhance performance, but this was only the case for one player [25]. In any case, nicotine consumption before and after sport can be hazardous, especially for those who are at higher risk considering their age, fitness level, or intensity of exercise. Combustible cigarette smoke and even nicotine alone impair oxygen demand/supply and increase circulating free fatty acids, which are associated with arrhythmic and pro-thrombotic events. This is particularly relevant considering that exercise alone evokes a similar phenomenon [9,15]. There are additional risks for bystanders, as second-hand smoke also increases CV risk, in particular in exposed children [4]. Finally, non-smokers may be more prone to take up smoking when they are influenced by behaviors of their peers, i.e., referees and coaches. Exposure to smoking peers has been shown to be a gateway to cigarette smoking, and coaches may have a major role as lifestyle models [18,26,27]. However, the influence of coaches remains controversial, as has been reported with smokeless tobacco in USA baseball players but not with combustible cigarette smoking in female basketball players [28,29,30].

Awareness of the risks associated with tobacco use was weak, both for the overall health risks and for the risks specific to consumption and the practice of sports. Almost half of respondents were unaware that there is an increased risk associated with tobacco use in the two hours before and/or following a sport session. Referees and coaches were not well informed (>35%, for both), which may represent a missed opportunity for health education amongst the players. In a large cohort of competitive athletes, Chevalier et al. found higher rates of awareness of the risks of smoking in the 2 h before or after sport (83% vs. 71% in our subjects) [16]. Risk awareness in smokers was associated with healthier behavior before and after practicing sport. However, despite this knowledge, more than 80% of the combustible cigarette smokers in our population smoked <2 h before or after a rugby session. Only few studies have addressed tobacco-related attitudes and knowledge in coaches [13,26,31,32,33,34]. Coaches are key stakeholders and role models in sport, and health education needs to be strengthened accordingly in this group [18,35,36]. Moreover, given the high rate and unhealthy use of e-cigarettes among youth sports participants, referees and coaches should be particularly aware of the effects of these devices [37].

### 4.1. Study Limitations

The validity of self-reports addressing tobacco consumption has been validated elsewhere, even if such behaviors could be underreported [38]. As respondent rate was quite low in players, extrapolation of our results to the amateur rugby players, coaches, and referees in general is only hypothetical. However, response rate was high among referees (almost 90%) and our results are in agreement with other work, especially regarding nicotine consumption in a sports setting and risk awareness. Cannabis consumption, included in the World Anti-Doping Agency’s list of prohibited substances, was not addressed in our study, because even if the questionnaire was anonymous, this issue might have negatively influenced the sincerity and rate of responses. Regional disparities on smoking use may limit the widespread extrapolation of our findings [3]. Nationwide surveys are therefore warranted to confirm our findings. Moreover, given the differences in age groups between rugby categories, which may affect smoking status, results on smoking habits should have been analyzed in sub-groups split for both age and categories. However, the low number of subjects did not allow for such a stratified analysis.

### 4.2. Perspectives

Participation in team sports should be strongly encouraged for the major social and health benefits that it provides, but, at the same time, there is an urgent need to ramp up smoking prevention campaigns among those involved [29]. Participating in a sport is usually associated with less smoking and less consumption of other addictive substances [21]. The protective effect of risk awareness, although incomplete, is a sign that improving the awareness of rugby players, coaches, and referees could influence positive behaviors. In addition to ongoing school programs, intervention in amateur sports clubs, if supported by a volunteer network, could provide the opportunity to enhance health education and promote smoking prevention [18,19,33,34,39,40]. Our findings may help to implement information sessions targeting coaches and referees about knowledge of the risk and behaviors. The educational and social roles of coaches and referees in health promotion were underlined, in particular as models regarding avoidance of combustible cigarettes and e-cigarettes.

## 5. Conclusions

Combustible cigarette smoking remains highly prevalent in amateur rugby in players as well as in coaches and referees, and the prevalence is not lower than in the general population of the same age. Persistent smoking rates in young female rugby players are particularly alarming. Moreover, the high rate of combustible cigarette smoking in the context of sport sessions is a high-risk attitude, not only for the smoker but also for the other players, referees, and coaches exposed to second-hand smoke. In addition, this use could further facilitate the initiation of non-smokers. Our results highlight the need for prevention programs to raise awareness regarding the risks associated with tobacco use to avoid cigarette smoking initiation and to encourage smoking cessation. We suggest referees and coaches as relevant targets for prevention and key levers of health education programs, in particular those involved in training female athletes.

## Figures and Tables

**Figure 1 ijerph-18-05720-f001:**
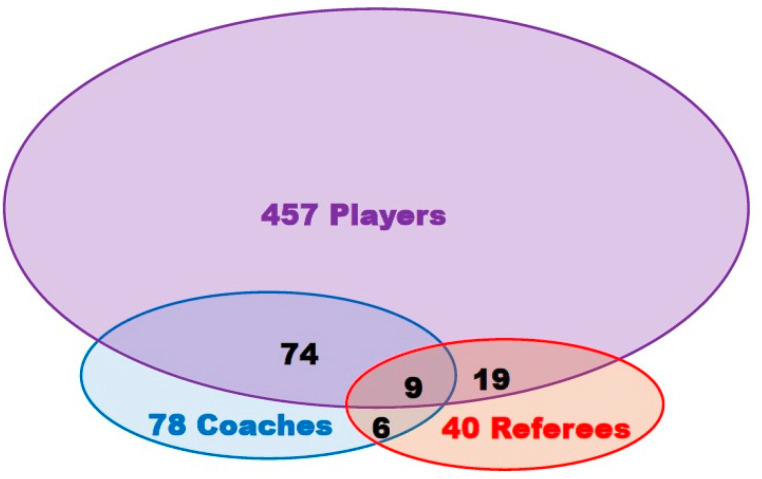
Number of respondents according to category of rugby practice and mutual interplay.

**Figure 2 ijerph-18-05720-f002:**
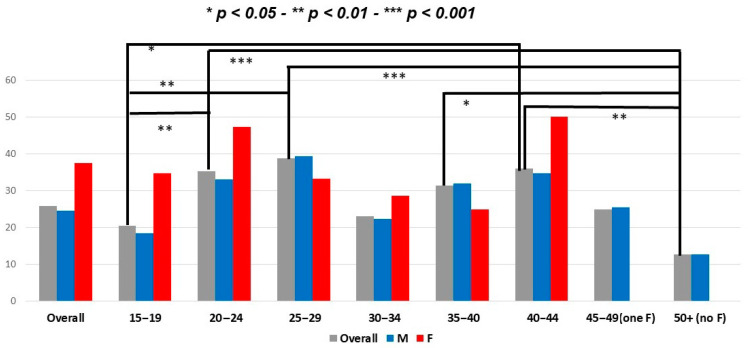
Rate of smokers in the whole study population and according to category of rugby practice and gender. M: male; F: female.

**Figure 3 ijerph-18-05720-f003:**
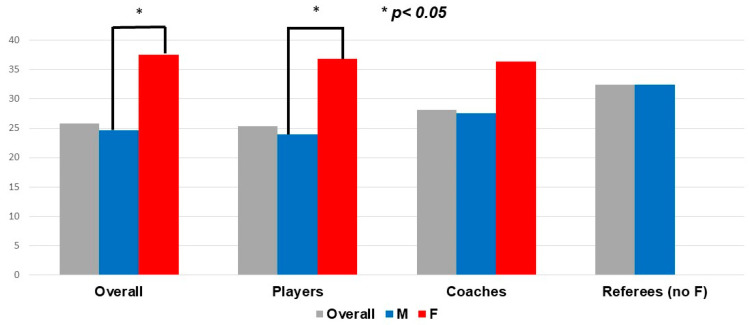
Rate of smokers in the whole study population and according to age group and gender. M: male; F: female. * *p* < 0.05. Grey = overall population; red = female; blue = male.

**Figure 4 ijerph-18-05720-f004:**
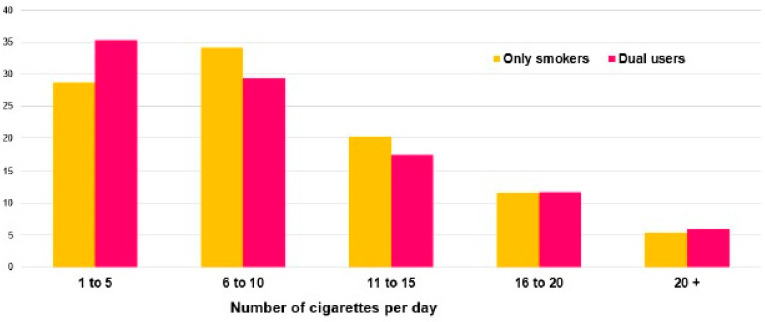
Level of daily combustible cigarette consumption in dual users (red bar) and those who only smoke combustible cigarettes (yellow bar) (number of cigarette/day).

**Figure 5 ijerph-18-05720-f005:**
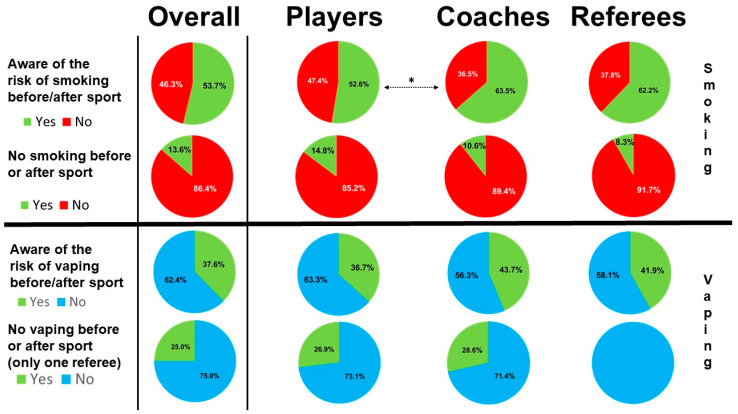
Level of knowledge on the risks of smoking (upper part) or vaping (lower part) in the setting of a rugby session (before/after) and related behaviors in the whole study population and according to category of rugby practice (* *p* < 0.05) Green color indicates a healthy attitude/behavior, and red or blue colors indicate an unhealthy attitude/behavior.

**Table 1 ijerph-18-05720-t001:** Characteristics of the study population in the whole population and according to the type of participant {n (%)}. Bold: categories and age groups.

	Overall		Players		Coaches			Referees	
**Population**	683		559 (81.8%)		167 (24.5%)			74 (10.8%)	
**Male**	619 (90.6%)		502 (89.8%)		156 (93.4%)			74 (100%)	
**Female**	64 (9.4%)		57 (10.2%)		11 (6.6%)			0 (0%)	
**Age group (y)**	**12–15**	**15–19**	**20–24**	**25–29**	**30–34**	**35–39**	**40–44**	**45–49**	**50+**
**Population**	31 (4.5%)	170 (24.9%)	119 (17.4%)	67 (9.8%)	65 (9.5%)	54 (7.9%)	50 (7.3%)	56 (8.2%)	71 (10.4%)
**Male**	31 (5.0%)	147 (23.7%)	100 (16.2%)	61 (9.9%)	58 (9.4%)	50 (8.1%)	46 (7.4%)	55 (8.9%)	71 (11.5%)
**Female**	0 (0%)	23 (35.9%)	19 (29.7%)	6 (9.4%)	7 (10.9%)	4 (6.3%)	4 (6.3%)	1 (1.6%)	0 (0%)

**Table 2 ijerph-18-05720-t002:** Characteristics of smokers (upper part) and vapers (lower part) in the whole population and according to the type of participant {*n* (%)}. Bold and shading: categories and age groups. NA: not applicable.

Smokers	Overall			Players	Coaches		Referees	
**Total**	176/683 (25.8%)			142/559 (25.4%)	47/167 (28.1%)		24/74 (32.4%)	
**Male**	152/619 (24.6%)			120/502 (23.9%)	43/156 (27.6%)		24/74 (32.4%)	
**Female**	24/64 (37.5%)			22/57 (38.6%)	4/11 (36.4%)		0/0 (NA)	
**Age group (year)**	**15–19**	**20–24**	**25–29**	**30–34**	**35–39**	**40–44**	**45–49**	**50+**
**Overall**	35/170 (20.6%)	42/119 (35.3%)	26/67 (38.8%)	15/65 (23.1%)	17/54 (31.5%)	18/50 (36.0%)	14/56 (25.0%)	9/71 (12.7%)
**Male**	27/147 (18.4%)	33/100 (33.0%)	24/61 (39.3%)	13/58 (22.4%)	16/50 (32.0%)	16/46 (34.8%)	14/55 (25;5%)	9/71 (12.7%)
**Female**	8/23 (34.8%)	9/19 (47.4%)	2/6 (33.3%)	2/7 (28.6%)	1/4 (25.0%)	2/4 (50%)	0/1 (0%)	0/0 (NA)
**Vapers**	**Overall**			**Players**	**Coaches**		**Referees**	
**Total**	28/683 (4.1%)			26/559 (4.7%)	7/167 (4.2%)		1/74 (1.4%)	
**Male**	24/619 (3.9%)			22/502 (4.4%)	7/156 (4.5%)		1/74 (1.4%)	
**Female**	4/64 (6.3%)			4/57 (7.0%)	0/11 (0%)		0/0 (NA)	
**Age group (year)**	**15–19**	**20–24**	**25–29**	**30–34**	**35–39**	**40–44**	**45–49**	**50+**
**Overall**	6/170 (3.5%)	6/119 (5.0%)	6/67 (9.0%)	3/65 (4.6%)	3/54 (5.6%)	1/50 (2%)	2/56 (3.6%)	1/71 (1.4%)
**Male**	3/147 (2.0%)	5/100 (5.0%)	6/61 (9.8%)	3/58 (5.2%)	3/50 (6.0%)	1/46 (2.2%)	2/55 (3.6%)	1/71 (1.4%)
**Female**	3/23 (13.0%)	1/19 (5.3%)	0/6 (0%)	0/7 (0%)	0/4 (0%)	0/4 (0%)	0/1 (0%)	NA

**Table 3 ijerph-18-05720-t003:** Characteristics of individuals who intended to quit smoking {n (%)}. e-cig: electronic cigarette. Bold and shading: categories and age group. NA: not applicable.

Intent to Quit	Overall		Players		Coaches		Referees	
**Overall**	109/176 (61.9%)		83/142 (58.5%)		31/47 (66.0%)		17/24 (70.8%)	
**With e-cigarette**	27/109 (24.8%)		25/83 (30.1%)		4/31 (12.9%)		5/17 (29.4%)	
**Male**	95/152 (62.5%)		70/120 (58.3%)		29/43 (67.4%)		17/24 (70.8%)	
**With e-cigarette**	23/95 (24.2%)		21/70 (30.0%)		4/29 (13.8%)		5/17 (29.4%)	
**Female**	14/24 (58.3%)		13/22 (59.1%)		2/4 (50.0%)		0/0 (NA)	
**With e-cigarette**	4/14 (28.6%)		4/13 (30.8%)		0/2 (0%)		0/0 (NA)	
**Age group (year)**	**15–19**	**20–24**	**25–29**	**30–34**	**35–39**	**40–44**	**45–49**	**50+**
**Overall**	17/35 (48.6%)	26/42 (61.9%)	16/26 (61.5%)	9/15 (60.0%)	14/17 (82.4%)	11/18 (61.1%)	9/14 (64.3%)	7/9 (77.8%)
**With e-cigarette**	6/17 (35.3%)	4/26 (15.4%)	6/16 (37.5%)	5/9 (55.6%)	4/14 (28.6%)	1/11 (9.1%)	1/9 (11.1%)	0/7 (0%)
**Male**	15/27 (55.6%)	20/33 (60.6%)	14/24 (58.3%)	7/13 (53.8%)	13/16 (81.3%)	10/16 (62.5%)	9/14 (64.3%)	7/9 (77.8%)
**With e-cigarette**	5/15 (33.3%)	4/20 (20.0%)	6/14 (42.9%)	3/7 (42.9%)	3/13 (23.1%)	1/10 (10%)	1/9 (11.1%)	0/7 (0%)
**Female**	2/8 (25.0%)	6/9 (66.7%)	2/2 (100%)	2/2 (100%)	1/1 (100%)	1/2 (50%)	NA	NA
**With e-cigarette**	1/2 (50.0%)	0/6 (0%)	0/2 (0%)	2/2 (100%)	1/1 (100%)	0/1 (0%)	NA	NA

**Table 4 ijerph-18-05720-t004:** Characteristics of dual users in the whole population and according to the type of participant, age group, and gender {n (%)}.Bold and shading: type of participant and age groups. NA = Not applicable.

Dual Users	Overall		Players		Coaches		Referees	
**Overall**	19/683 (2.8%)		19/559 (3.4%)		4/167 (2.4%)		1/74 (1.4%)	
**Male**	17/619 (2.7%)		17/502 (3.4%)		4/156 (2.6%)		1/74 (1.4%)	
**Female**	2/64 (3.1%)		2/57 (3.5%)		0/11 (0%)		0/0 (NA)	
**Age group (year)**	**15–19**	**20–24**	**25–29**	**30–34**	**35–39**	**40–44**	**45–49**	**50+**
**Overall**	5/170 (2.9%)	5/119 (4.2%)	5/67 (7.5%)	1/65 (1.5%)	2/54 (3.7%)	0/50 (0%)	1/56 (1.8%)	0/71 (0%)
**Male**	3/147 (2.0%)	5/100 (5.0%)	5/61 (8.2%)	1/58 (1.7%)	2/50 (4.0%)	0/46 (0%)	1/55 (1.8%)	0/71 (0%)
**Female**	2/23 (8.7%)	0/19 (0%)	0/6 (0%)	0/7 (0%)	0/4 (0%)	0/4 (0%)	0/1 (0%)	NA

## Data Availability

Not applicable.

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
