# Peer review of "Smoking and Vaping in Amateur Rugby Players, Coaches and Referees: Findings from a Regional Survey Might Help to Define Prevention Targets"

_ijerph, 2021, doi:10.3390/ijerph18115720_

Round 1
Reviewer 1 Report
- The authors are encouraged to refer to combustible cigarettes as tobacco products and e-cigarette use as ENDS. It will be more accurate to such phrasing and congruent with tobacco control field, internationally. For instance, authors use “analyzed tobacco consumption’’; however, they analyzed both tobacco smoking and nicotine use.
- It is not necessary to minimize that 4% of smokers were dual users. It is in line with evidence about vaping in the context of dual use, particularly in settings where smoke-free policies are more relaxed.
- Please consider adding 2-3 sentences about the actual rates of smoking and vaping in France, particularly among the age group of the athletes and comparable sub-groups. There is some mention of those in the discussion; however, those should be at introduction to help the reader contextualize findings.
- Please consider adding definitions of amateur rugby players (for instance, add highest end of age range; overall gender breakdown) and “regional scale” (for instance, define geographic area, a selected number of communities vs all of Burgundy, high level demographics). Moreover, what does it mean that the survey is contemporary?
- The authors are advised to add more information about human subjects protections, particularly those related to involving minors in research.
- Please consider reporting items in the questionnaire with more description (e.g., points on a scale, prior validated items with source).
- Regarding recruitment, how many links were sent out? What was the response rate? What percentage of rugby players in the region as a catchment area does that represent?
- Use of parentheses within parentheses makes it difficult to discern findings. Authors are encouraged to utilize brackets within parentheses.
- Findings related to females having higher rates of tobacco use are interesting, authors are encouraged to provide more information on this.
- It is interesting that some participants played multiple roles. Did the authors examine if participants who wore multiple hats had differences in their tobacco use?
- It is challenging to appreciate how age groups reported are related to the roles participants filled. I authors are encouraged to add a breakdown of age groups according to role alongside smoking/vaping.
- Regarding paragraph that begins with, “We also found that more than one third of participants had experience with e-cigarettes,” it appears that there are substantive data about e-cigarette use that will be more effectively placed at introduction.
- Authors discuss findings related to motivation to quit; however, there is no mention of that framework in introduction. That should be addressed early in paper. Additionally, authors are encouraged to have more definition to this measure as suggested earlier in Point #6.
- According to the title, the study is to identify opportunity to promote the prevention of tobacco/nicotine use. However, the manuscript speaks more to opportunity to promote smoking cessation. The authors can strengthen paper by adding more context to framework that guides this research.
- Regarding tables, there is opportunity to provide more details as to what some of the figures mean. For instance, the use of Total is not clear at a glance. It is more common to have an Overall number that aggregates the adjacent columns. The authors are encouraged to evaluate how the tables can include more Notes with definitions to help them be more independent.
- Overall, the use of certain language in the report can be more objectively presented. There is frequent use of language that can be perceived as value-laden and less objective when reported in English.
- It appears that there are many issues with formatting as requested by IJERPH. I suggest that author review journal requirements.
Reviewer 2 Report
This study aimed at significant public health problems.
However, this study requires some revisions:
- The introduction is too short - please provide one paragraph with a brief explanation of why do the authors decided to run this study among rugby players.
- Methods
- the authors should provide information how many rugby players, coaches and referees were invited - these 3 groups have different age so it would be important to know it;
- moreover, the authors should provide a brief description of questions that were used to assess smoking status (e.g. past 30-days use or other?)
- data on how the participants were encouraged to take the survey should be included (supplementary data are not visible for the reviewer)
3. Results
- in the reviewer opinion results (smoking habits) should be analyzed separately for players, coaches and referees - these 3 groups have different age and social role, so the smoking habits may be shaped by the abovementioned factors
- statistics used by the authors are very simple - please consider more advanced statistical methods e.g. logistic regression models
4. Discussion
- please clearly define the practical implications of this study
Round 2
Reviewer 1 Report
This is an interesting study with valuable findings. The value is in reporting tobacco and nicotine use among sub-groups. It is unfortunate that there is not a guiding framework on eliciting/examining risk perception and/or appraising harms of tobacco/nicotine, which has been extensively reported. However, its absence does not dissuade me from recommending this paper for publication.
Author Response
In order to answer to this remark, we attached a translated study questionnaire in supplementary material, as a framework on how we assessed risk perception and appraised knowledge on harms of tobacco and nicotine.
Reviewer 2 Report
The authors addressed most of the comments:
- Please attach a study questionnaire as a supplementary material
- "Smoking was defined as current cigarette smoking or stop smoking < 3 months" - the authors should precise "smoking status was defined...."
Comments related to the presentation of the results are well-addressed by the authors.
Author Response
As requested, we attached a translated study questionnaire as a supplementary material; moreover, smoking status was defined as it was suggested.